# A Study on the Optimization of the Conductive Ball Manufacturing Process, Used for Anisotropic Conductive Films

Jong-Keun Choi, Young-Gyun Kim and Kwan-Young Han *

Department of Electronic and Electrical Engineering, Dankook University,
152, Jukjeon-ro, Suji-gu, Yongin-si 16890, Korea
* Correspondence: kyhan@dankook.ac.kr

**Abstract:** Currently, as the next-generation of display progresses—with high performance and high integration—the surface mounting technology of components is very important. In particular, in the case of flexible displays, such as rollable and bendable displays, ACF that connects wires to any curvature is essential. However, the conductive ball used inside the ACF has had problems with particle size and non-uniform metal coating. It was confirmed that the presence of solvent and oxygen, which are used in polymer synthesis, affects the sphere formation of polymer beads. By optimizing the factors affecting the polymer beads, a perfect spherical polymer bead was manufactured. In addition, the conductive ball manufacturing process was optimized by confirming the factors affecting the metal coating. The metal coating on the surface of the polymer bead was applied with a uniform thickness by considering the specific surface area and concentration of the conductive balls, and, through this optimized process, conductive balls for anisotropic conductive films with uniform size and metal thickness were obtained.

**Keywords:** anisotropic conductive film (ACF); polymer bead; conductive ball; metal coating

## 1. Introduction

Recently, flat panel displays, such as TVs, smartphones, tablets, and advertisement panels, are rapidly being replaced by flexible displays, regardless of the display size. The core technology that satisfies these changes is the fine-pitch electronic packaging technology. Although ball grid array (BGA) packaging technology is used to connect conductors in the existing display, it has a disadvantage in that the area of the substrate is large and it is impossible to use it on a flexible substrate [1–4]. When connecting conductors in a flat panel display, ball grid array (BGA) packaging technology is used, but it has disadvantages in that it is impossible to simplify the bezel and use a flexible substrate. To overcome this problem, most companies are using anisotropic conductive film (ACF). ACF is a film type in which conductive balls—made by coating nickel and gold on a polymer surface for mechanical connection and vertical (z-direction) transmission inside—are dispersed in an adhesive. The packaging method using ACF satisfies the decreasing thickness and reliability of the panel and has the advantage of improving the mounting density and realizing an ultra-small or ultra-thin display [5]. However, a terminal connection technology of a fine-pitch electrode and a driver IC is required for high resolution. In addition, as the width of the pitch decreases, it becomes increasingly difficult to catch the conductive balls present in the ACF [6–10]. These problems are caused by particle size and non-uniform metallic coating, which can affect the mechanical properties and reliability. In this study, a conductive ball with an optimized particle size was fabricated to lower the contact resistance at a fine pitch. First, a polymer bead was prepared to control the size of the conductive ball. Particles are created through radical polymerization, and their size and spherical shape are modified by external factors such as oxygen and moisture [11,12]. In addition, the uniformity and thickness of the metal coating during the metal coating process are affected by factors such

as the specific surface area and the plating solution concentration, thereby affecting the contact resistance [13–16].

## 2. Materials and Methods

### 2.1. Reagents and Materials

Styrene (St), butyl acrylate (BA), methacrylic acid (MAA), and 4,4′-azobis (4-cyanovaleric acid) (ACVA) were purchased from Sigma-Aldrich, Korea (Seoul, Korea), and the buffers—tetrahydrofuran (THF), aluminum oxide, and sea sand—were purchased from Samjeon Soonyak (Seoul, Korea). Styrene (St) and butyl acrylate (BA) were mixed with tetrahydrofuran (THF) in a 1:1 ratio, and aluminum oxide and sea sand were layered on the column, followed by filtering and purification. Additives mixed in the material were removed through filtering, and the THF mixed in the material was removed using a concentrator. Finally, the remaining THF was removed using a vacuum chamber, and the purified material was used. The buffers, methacrylic acid (MAA) and ACVA were used without further purification.

### 2.2. Pre-Treatment

First, to purify styrene (St) and butyl acrylate (BA), they were mixed with organic solvent THF in a ratio of 1:1. The reason for mixing tetrahydrofuran (THF) was to obtain 100% purity, by decomposing the additive in which St and BA are mixed. Next, aluminum oxide and sea sand were stacked in a column in a ratio of 9:1, followed by filtration and purification. Additives mixed in St and BA were removed through a filter, and THF mixed in the material was removed using a concentrator. At this time, the temperature of the chiller connected to the concentrator was −5 °C and the RPM was 250 for 1 h. Finally, the remaining THF was removed using a vacuum chamber to obtain purified St and BA. Buffers, methacrylic acid (MAA), and ACVA were used without further purification [17].

### 2.3. Synthesis of Poly (Styrene–Butyl Acrylate–Methacrylic Acid) Seed Microspheres

The radical polymerization reaction of poly(St-BA-MAA) was synthesized through chemical bonding, as shown in Figure 1. Nitrogen was put into a two-necked, round-bottomed flask to form an atmosphere. A buffer (100 mL) was added, and St (8 mL), BA (100 μL), MAA (120 μL), and ACVA (0.04 g) were successively added to the flask. The reaction was stirred at 350 rpm. After stirring for 15 min, the reagent mixture was heated to 75 °C in an oil bath, for 8 h.

**Figure 1.** Chemical synthesis to fabricate polymer beads.

### 2.4. Radical Polymerization Mechanism

As a method of manufacturing polymer beads, radical polymerization is generally used. The radical polymerization mechanism can be explained by dividing it into the following four stages: initiation, propagation, termination, and chain transfer. First, initiation is the stage in which the initiator forms a radical and then the radical reacts with the monomer. Figure 2, below, shows the chemical structure of the initiator decomposition reaction and the initiation reaction [18,19].

**Figure 2.** Radical mechanism initiation: (**a**) initiator decomposition and (**b**) initiation reaction.

Next, the propagation is a reaction in which monomers continue to bond and the chain length increases to form a polymer. As shown in Figure 3, it can be seen that the monomer continues to increase with time.

**Figure 3.** Radical mechanism propagation.

Third, the termination is a state in which two radicals react and the radicals disappear. Figure 4, below, shows the chemical structure when two reactions occur: a bonding reaction and a heterogeneous reaction.

**Figure 4.** Radical mechanism termination: (**a**) combination and (**b**) disproportionation.

Finally, the transfer is a reaction in which a growing polymer radical reacts with a molecule to form a polymer and a radical molecule is generated. In this case, the molecule is also called a chain transfer agent. Based on this mechanism, the reason why oxygen should be removed can be explained. Figure 5, below, is the chemical structure showing the presence of oxygen in radical polymerization.

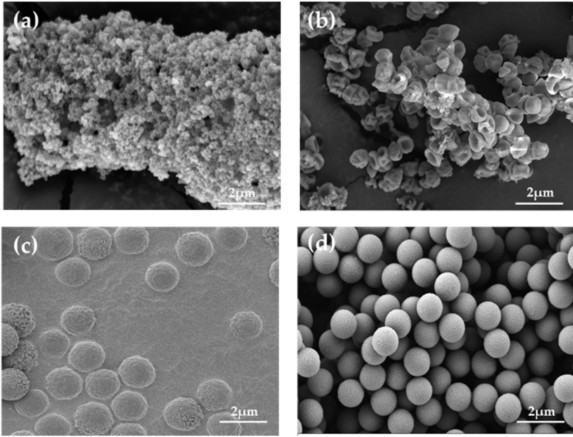

**Figure 5.** Chemical structure that appears in the presence of oxygen during radical reactions.

It can be easily understood by comparing it with the second propagation reaction in the mechanism, divided into four steps, described above. Looking at Figure 3, it can be seen that the chain length increases as the monomer continues to bond when radicals are generated. However, when oxygen is present, it reacts with oxygen first to form a copolymer with oxygen. For this reason, oxygen must be removed so that radicals can react with monomers to form polymers. To remove oxygen, a degassing process was performed from inside.

### 2.5. Measurements

Gas chromatography (PerkinElmer, Waltham, MA, USA) was used to check the presence of THF inside the purified material. The surface and shape of the manufactured conductive ball were observed using a scanning electron microscope (SEM, HITACHI, Tokyo, Japan). In addition, energy dispersive spectrometer (EDS) was used to confirm the metal coating on the surface.

## 3. Results and Discussion
### 3.1. Factors Affecting Polymer Beads

In the process of preparing the material, the material was purified by mixing it with THF to remove additives and impurities, to prepare the polymer beads. Figure 6 shows the change in the shape of the beads over time between the concentrator and the low-pressure chamber, using the characteristics of the THF—a volatile solution that vaporizes even at low temperatures. In each image, the amount of THF was converted into a numerical value using gas chromatography, based on the time to purify the material.

**Figure 6.** SEM image of polymer shape, depending on the amount of THF remaining in the material: (**a**) composite image of material with THF 9.64 wt%, (**b**) composite image of material with THF 3.15 wt%, (**c**) composite image of material with THF 0.03 wt%, and (**d**) composite image of material with THF 0 wt%.

As shown in Figure 6a, it was confirmed that the material with the amount of 9.64 wt%, to a material having a small amount of THF of 0.03 wt%, had an effect on the synthesis of the polymer beads. Although it is small, depending on the amount of THF, the reason there is an effect is because it has the property of dissolving organic solvents. Due to this property, polymer synthesis was hindered, and the shape of the sphere was not properly formed.

When the THF was completely removed, it can be observed that a proper sphere shape was formed, as shown in Figure 6d. Based on these results, it is believed that a purification process of at least 4 h is necessary to prevent the THF, which is mixed for purification, from remaining in the material. A polymer was synthesized by radical polymerization, using the optimized material and an initiator that was purified as described above. At this time, an internal atmosphere was maintained using nitrogen, but radical synthesis may not have occurred properly due to oxygen mixed in the nitrogen.

Figure 7 shows the formation of polymers, based on the presence or absence of oxygen. Figure 7a shows the appearance of a polymer when oxygen was maintained without maintaining a nitrogen atmosphere. It can be observed that the polymer was not synthesized properly and has a gum-like shape, not a spherical shape. The reason for this result is because there was an odd number of electrons, which met with the oxygen in an unstable state and caused a reaction, which resulted in a polymer synthesis that was not performed properly. To solve this problem, the experiment was conducted by maintaining a nitrogen atmosphere rather than an oxygen atmosphere and by installing a trap to remove both oxygen and moisture from the outside or inside the nitrogen. As shown in Figure 7b, it can be observed that the polymer is spherical.

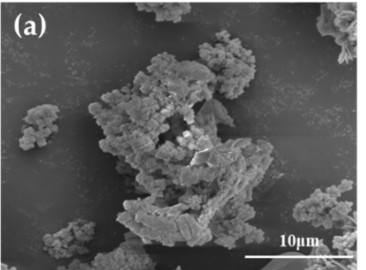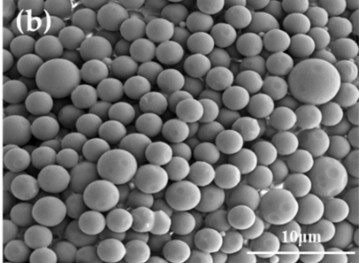

**Figure 7.** SEM image of polymer shape with and without oxygen: (**a**) composite image of polymer exposed to oxygen, and (**b**) composite image of polymer produced by blocking oxygen.

### 3.2. Surface Modification and Cleaning of Fabricated Polymer Beads

Surface modification was conducted to apply a metal coating onto the surface of the polymer bead, prepared as described above.

Figure 8 is an image of the polymer surface, according to each surface treatment. Although there is no significant difference between the SEM images, it was observed that the polymer beads had a white surface until cleaning, but that the surface color changed to red after the surface modification, and then to gray after the last catalytic treatment. First, water washing was performed using DI water to remove the buffer remaining on the polymer beads. Next, to remove the foreign substances or chemicals remaining on the surface of the polymer, cleaning and conditioning were performed, followed by washing with water. After removing the foreign substances from the surface, activation, which is a surface modification solution that is used to cause a reaction with the metal ions, was performed on the polymer surface, followed by washing with water again. Finally, acceleration, which serves as a catalyst to increase the reaction rate of the surface coating, was coated and then washed with water to finish the surface modification of the polymer.

### 3.3. Optimization of Metal Coating on the Surface of the Fabricated Polymer Bead

A nickel-plating solution was prepared to coat the nickel metal on the surface-modified polymer beads. At this time, the plating salt dropping method was used. The plating salt dropping method, is a method in which polymer beads are put into a reducing agent solution and stirred, and then a nickel ion solution is added dropwise, at a constant rate to cause a reaction. Next, because the pH of the reducing agent solution changes according to the amount of the nickel ion solution, the application of the metal coating was conducted while checking the pH meter, to maintain the pH. After the reaction was completed, I was able to get a conductive ball through defense.

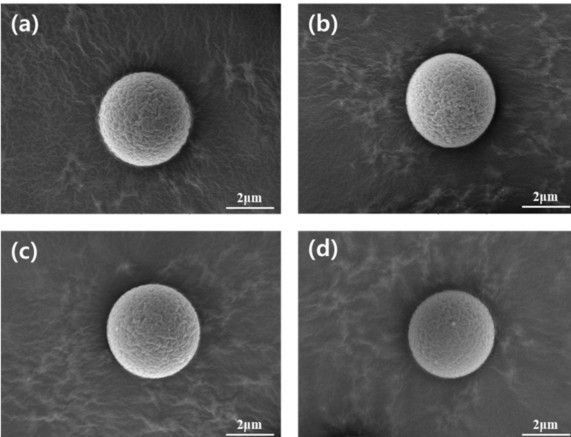

**Figure 8.** SEM image, according to polymer surface treatment: (**a**) polymer surface image after water washing, (**b**) surface image after cleaning and conditioning treatment, (**c**) surface image after activation treatment, and (**d**) surface image after acceleration treatment.

Figure 9 shows the surface-treated polymer beads, coated with nickel. It can be seen that most of the beads are not only aggregated, but also overcoated. The reason for this is because the nickel solution enters excessively, compared to the specific surface area of the polymer beads, and an overload occurs. The specific surface area of the polymer bead can be obtained as shown in Equation (1), below.

$$Specific\ surface\ area\left(m^2/g\right) = \frac{1\ spherical\ area}{1\ sphere\ mass} = \frac{4\pi r^2}{\rho \frac{4}{3}\pi r^3} = \frac{3}{r\rho} \tag{1}$$

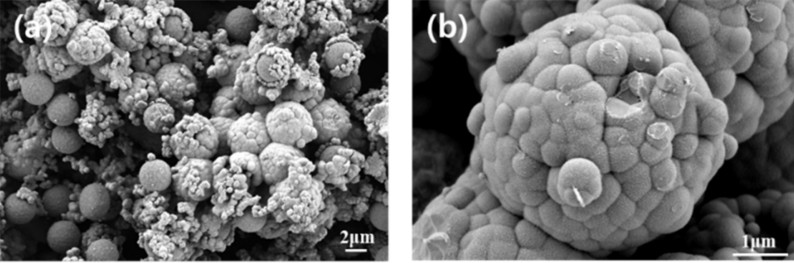

**Figure 9.** SEM image of polymer beads coated with metal: (**a**) image taken at 8.0 K magnification and (**b**) image taken at 40.0 K magnification.

Here, $r$ denotes the radius of the sphere and $\rho$ denotes the specific gravity. The amount of the metal coating solution can be adjusted by calculating the specific surface area of the polymer bead, prepared using the above formula.

Figure 10 is an image of a conductive ball that was coated with nickel after calculating the specific surface area and is a cross-sectional view of the conductive ball, using FIB SEM. Figure 10a,b, demonstrate that the overall coating is better, compared with that in Figure 9. Figure 10c is a graph analyzing the components of the conductive balls, using EDS. Looking at the peak values in the graph for the carbon atoms and oxygen atoms, these peak values are attributed to the carbon tape used to fix the conductive balls. For the phosphorus (P) atoms, the peak value corresponds to the element that was used when coating nickel on the polymer surface. Equation (2), below, shows the chemical formula for coating nickel on the polymer surface.

$$NiSO_4 + 2NaH_2PO_2 \rightarrow Ni + 2NaH_2PO_3 + H_2 + H_2SO_4$$
$$NaH_2PO_2 + H \rightarrow H_2O + NaOH + P \tag{2}$$
$$Ni - P$$

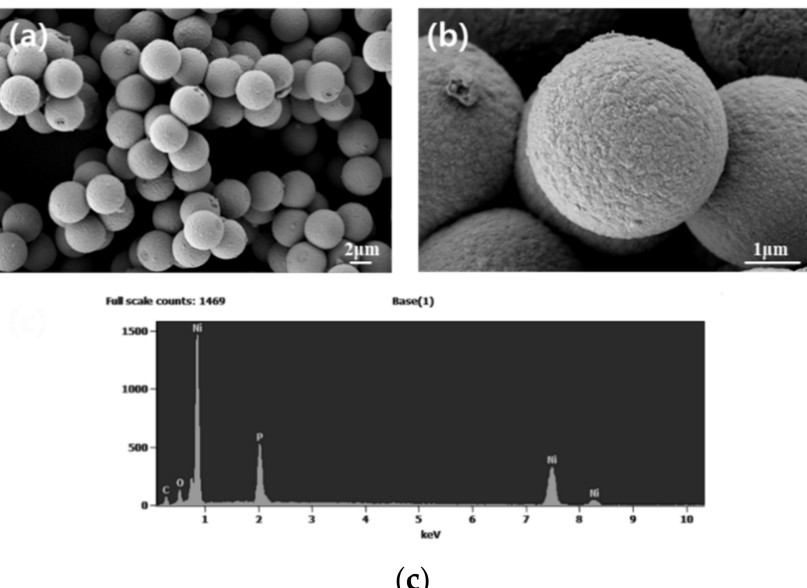

**Figure 10.** SEM image and EDS graph of metal coating, with optimizing concentration after calculating the specific surface area: (**a**) image taken at 8.0 K magnification, (**b**) image taken at 40.0 K magnification, and (**c**) surface composition analysis graph, using EDS.

It can be observed that nickel and phosphorus (P) are produced through the above chemical formula. The results confirm that the nickel coating was formed on the polymer surface through the significant increase in the peak values of the nickel and phosphorus, in Figure 10c.

Furthermore, from Figure 11a,b, it can be observed that the nickel was coated on the surface of the polymer with an overall even thickness. Through these results, it was found that if the quantity of the metal coating solution was calculated using the specific surface area, it was possible to evenly coat the polymer surface. A conductive ball was manufactured by coating metal on a polymer, using the plating drop method described above. It was previously confirmed that the results, shown in Figure 10, can be obtained when the specific surface area is calculated and a metal coating solution that is suitable for the quantity is added. However, the pH of the solution varies depending on the amount of metal coating solution and a change in the shape of the conductive ball can be observed, as shown in Figure 12, according to the value of the pH.

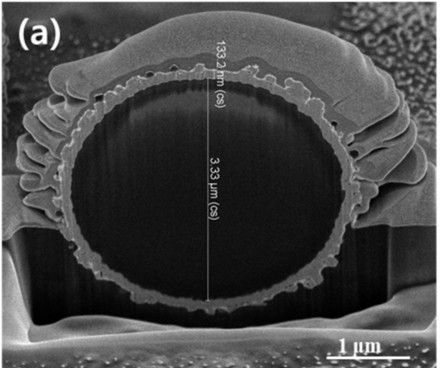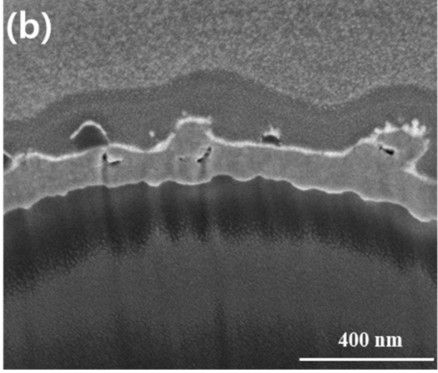

**Figure 11.** Metal-coated polymer, cross-sectional image: (**a**) image taken at 25 K magnification and (**b**) image taken at 100 K magnification.

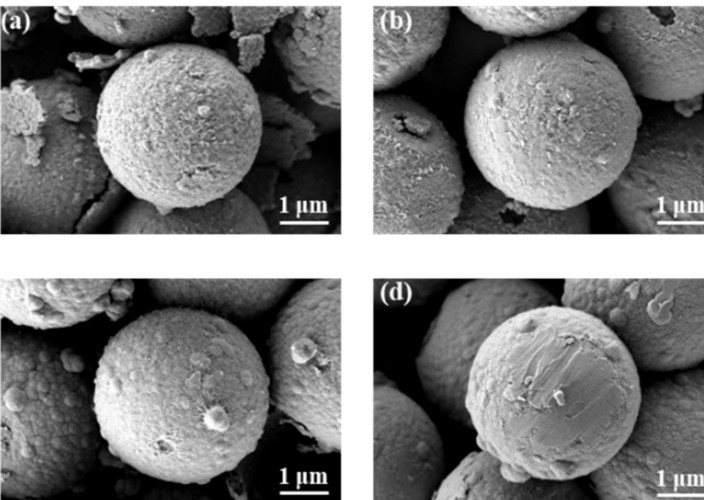

**Figure 12.** Image of metal coating surface changed according to pH: (**a**) image of conductive ball coated with metal at pH 4.2, (**b**) image of conductive ball coated with metal at pH 4.7, (**c**) image of conductive ball surface with metal coating at pH 5.2, and (**d**) image of conductive ball with metal coating at pH 5.7.

The roughness of the conductive ball surface changed as the pH value increased. With a pH of five or higher, the surface became rough, and crumbs were formed. In addition, it was found that the surface of the conductive ball itself was clean at a pH of 4.2, but nickel flakes were formed. Through these results, it can be concluded that a conductive ball with the best surface was produced when the pH was 4.7.

Finally, Figure 13 shows how the conductive ball surface changed according to the reaction time. When the reaction time was measured from 10 min to a maximum of 40 min, there was no significant change.

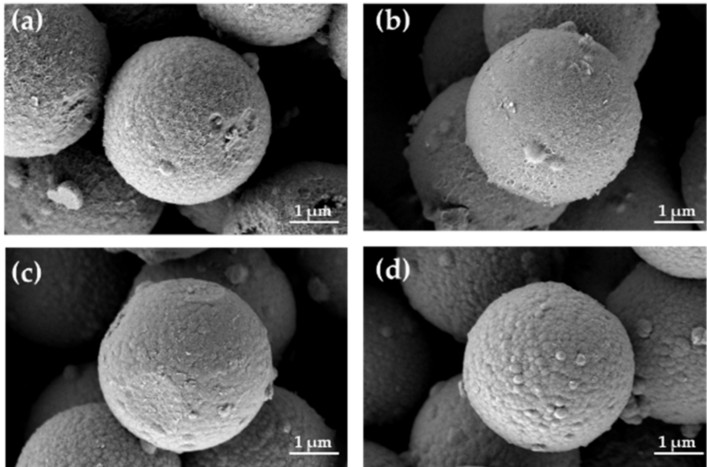

**Figure 13.** Image of conductive ball surface, changed according to metal coating reaction time: (**a**) surface image of conductive ball after 10 min of reaction time, (**b**) surface image of conductive ball after 20 min of reaction time, (**c**) surface image of conductive ball after 30 min of reaction time, and (**d**) surface image of conductive ball after 40 min of reaction time.

In conclusion, by adjusting the polymer ratio while optimizing the above conditions, polymers of various sizes were fabricated, as shown in Figure 14. It can be freely controlled from as small as 1 µm, to as large as approximately 3 µm. Through this size control, it is possible to optimize the size of the conductive ball so that it is suitable for the fine pitch used in next-generation displays.

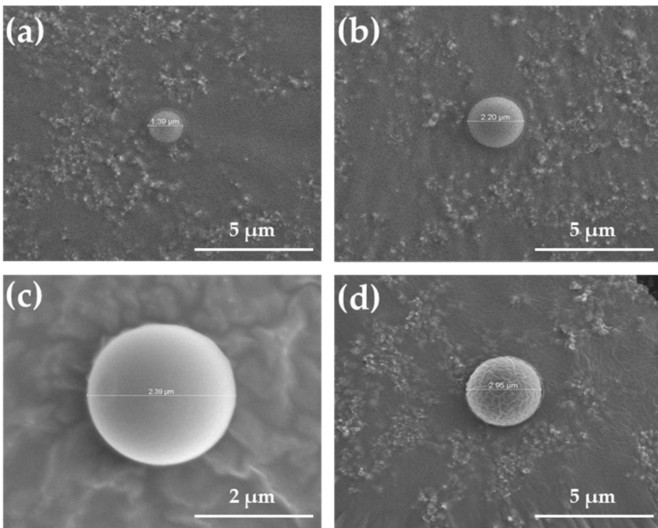

**Figure 14.** Polymer beads of various sizes, manufactured by an optimized process: (**a**) 1.33 μm bead, (**b**) 2.2 μm bead, (**c**) 2.39 μm bead, and (**d**) 2.95 μm bead.

Figure 15 shows that a flexible printed circuit board (FPCB), with a pitch distance of 20 μm, was fabricated to realize the fine pitch used in the next-generation displays. Figure 15a is an actual image of the fabricated FPCB. Figure 15b is a schematic diagram for implementing flex-on-glass (FOG). After making conductive balls of 1 μm, 2 μm and 3 μm in ACF form, pre-bonding was performed on the FPCB, at 80 °C 1 Mpa, for 2 s. Next, we pressed indium tin oxide (ITO)–glass at 200 °C and at 3Mpa pressure, for 2 s on the ACF and attached it to the FPCB.

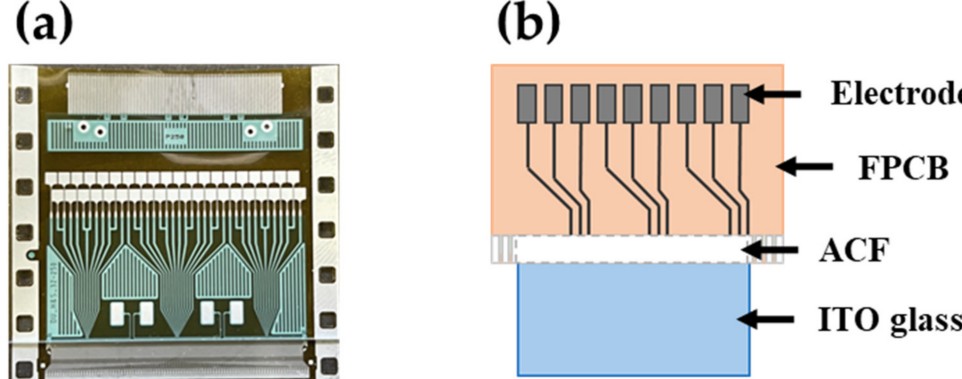

**Figure 15.** FPCB and schematic diagram for measuring the contact resistance of the manufactured conductive ball: (**a**) real sample of PFCB substrate with 20 μm pitch and (**b**) schematic diagram for composing flex-on-glass (FOG).

Figure 16 is a graph measuring the contact resistance, according to the size of the conductive ball. Com-3 μm is a commercial product and is a 3 μm-sized, gold-coated conductive ball currently in use. The conductive ball was manufactured by coating gold on the surface of 1 μm, 2 μm, and 3 μm-sized beads. The contact resistance of commercially available products have a size of 3 μm and the manufactured conductive ball is approximately 2 mΩ. However, the 1 μm-sized conductive ball resistance has a high resistance of 4 mΩ. The reason for such a result is that the conductive balls are too small, so more conductive balls must be included in the same volume, and then agglomeration occurs. A conductive ball with a size of 2 μm has a resistance that is about 0.5 mΩ lower than that of the conventional 3 μm size. As the pitch becomes narrower, the bump on the substrate is also reduced. When the conventional 3 μm conductive balls are used, the number of conductive balls in contact with the bumps is significantly reduced. Therefore, it is judged

that smaller conductive balls have lower resistance because they are more suitable for circuit bumps.

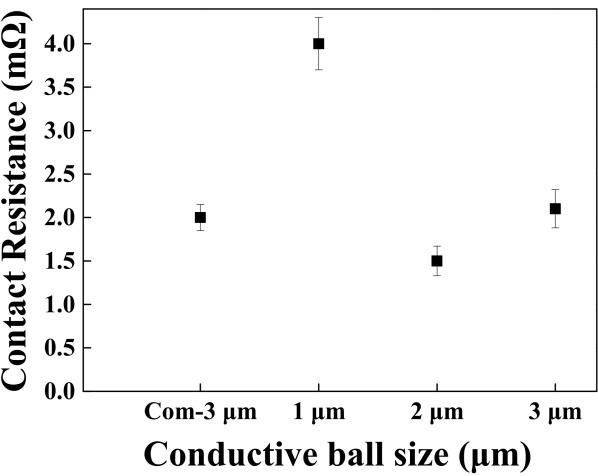

**Figure 16.** Contact resistance measurement graph, according to the size of the conductive ball.

## 4. Conclusions

In this paper, we propose a method for the fabrication of polymer beads for ACF, as well as the optimization of the metal coating process. It was confirmed that the shape of the polymer changes depending on the solvent and oxygen, affecting polymer production. It was found that polymer beads are not properly formed due to the radical reaction, which may not occur properly depending on the presence of the THF solvent and the presence or absence of oxygen. By solving these problems, we achieved a spherical shape of the polymer beads, which was a critical issue in the past. Additionally, the factors that could affect the metal coating were determined and optimized. By calculating the area of the polymer surface, based on the specific surface area and by determining a suitable concentration with this calculation, a conductive ball with a constant thickness was produced. Through these results, it was possible to manufacture beads of various sizes by optimizing the factors affecting the bead and metal coating, and, at the same time, it was possible to confirm a conductive ball that was suitable for a fine pitch.

**Author Contributions:** Conceptualization, J.-K.C., Y.-G.K. and K.-Y.H.; methodology, J.-K.C. and K.-Y.H.; formal analysis, J.-K.C. and Y.-G.K.; investigation, J.-K.C. and Y.-G.K.; resources, J.-K.C. and K.-Y.H.; data curation, J.-K.C. and K.-Y.H.; writing—original draft preparation, J.-K.C. and Y.-G.K.; writing—review and editing, J.-K.C., Y.-G.K. and K.-Y.H.; visualization, J.-K.C. and K.-Y.H.; supervision, K.-Y.H.; project administration, J.-K.C. All authors have read and agreed to the published version of the manuscript.

**Funding:** This research received no external funding.

**Institutional Review Board Statement:** Not applicable.

**Informed Consent Statement:** Not applicable.

**Data Availability Statement:** The data presented in this study are available on request from the corresponding author.

**Conflicts of Interest:** The authors declare that they have no conflict of interest.

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
