# Peer review of "A Study on the Optimization of the Conductive Ball Manufacturing Process, Used for Anisotropic Conductive Films"

_electronicmat, doi:10.3390/electronicmat3030021_

Round 1
Reviewer 1 Report
The authors reported the synthesis of conductive balls used for anisotropic conductive films, and their post-treatments based on different strategies (i.e., water washing, cleaning and conditioning treatment, activation, and acceleration treatment). Though, it is not so clear how these methods were performed. This is an interesting work. However, some statements could be clarified for the readers. Thus, the manuscript should be improved by addressing the following comments.
1. The country of the University is not mentioned in the affiliation.
2. Abstract: The sentence “The ACF(anisotropic conductive film) used to connect the wires between electronic prod-8 ucts is essential” looks incomplete. From what point of view is this important?
3. The Introduction section is too poor. A literature review needs to be done and presented. Authors should show the properties, types of materials, and applications of ACFs, for example.
4. The name of the THF solvent is not mentioned in the manuscript. Please review the Experimental section and add all the equipment and reagents brands.
5. It is not so clear how the post-treatment methods were performed. Include a subsection in the Experimental section, detailing each method. Also, were the methods based on other works of the literature? If yes, please, include the references.
6. The quality of the reaction in Figure 1 should be improved.
7. Figure 2: The mention and discussion about the SEM images need to be changed to the Results section, instead of the Experimental section. Please, inform the magnification and the scale of all SEM images.
8. Include references to support the findings in the results section. The comparison with other reported materials is also interesting and strongly recommended.
9. In the sentence “Figure 5 is a photograph of the surface-treated polymer beads coated with nickel metal.”, please confirm the term “photograph”, it seems to be a SEM image.
10. What is the average size/diameter of the conducting balls? Are there changes before and after treatments? And after the metal coating?
11. Author said that “There was no significant difference compared with the commercially available products, and the contact resistance of the conductive balls” and “…there was no significant difference from the commercial products when they were coated with nickel or gold.” However, no information or SEM image of the commercial material is shown. I suggest that authors include these results performed with the commercial material, as well as the resistance or other properties informed by the manufacturer. Maybe the comparison between both materials could be shown in a table.
Author Response
to the reviewer
Thanks for the advice you pointed out.
I felt that there were a lot of things that needed to be corrected through the advice.
I added and corrected the missing data by referring to the advice.
The revised thesis will be sent again with attached files.
If you click Review in Word to check, you can check the revised part.
Thanks again.
-------------------------------------------------------------------------------------
1. The country of the University is not mentioned in the affiliation.
-> I have edited the part you mentioned.
2. Abstract: The sentence “The ACF(anisotropic conductive film) used to connect the wires between electronic prod-8 ucts is essential” looks incomplete. From what point of view is this important?
-> ACF is essential for flexible displays such as rollable and bendable displays and electronic products with curvature. This is because, in the case of solderable, which is an existing surface mount technology, cracks occur when curvature occurs and the connection between electronic products is broken
3. The Introduction section is too poor. A literature review needs to be done and presented. Authors should show the properties, types of materials, and applications of ACFs, for example.
-> The introduction was newly revised, and insufficient data was added.
4. The name of the THF solvent is not mentioned in the manuscript. Please review the Experimental section and add all the equipment and reagents brands.
-> Added equipment and reagent brands in the experimental section.
5. It is not so clear how the post-treatment methods were performed. Include a subsection in the Experimental section, detailing each method. Also, were the methods based on other works of the literature? If yes, please, include the references.
->Materials and methods section corrected. Each part explained in detail how to do it. Added a reference. (R. vijayaraghavan, J. M. Pringle, D. R. MacFarlane, European Polymer Journal, 1758-1762, 44 (2008))
6. The quality of the reaction in Figure 1 should be improved.
-> Changed to a picture with improved quality
7. Figure 2: The mention and discussion about the SEM images need to be changed to the Results section, instead of the Experimental section. Please, inform the magnification and the scale of all SEM images.
-> Changed the contents of Figure 2 to the Results section. In addition, the entire experimental section has been revised. All the magnifications of the SEM images were filled in.
8. Include references to support the findings in the results section. The comparison with other reported materials is also interesting and strongly recommended.
-> The result of fine pitch when applied to the existing 3um size and the result when the 2um size was produced were added and compared.
9. In the sentence “Figure 5 is a photograph of the surface-treated polymer beads coated with nickel metal.”, please confirm the term “photograph”, it seems to be a SEM image.
-> The photograph has been deleted.
10. What is the average size/diameter of the conducting balls? Are there changes before and after treatments? And after the metal coating?
-> There is no change in size before and after surface treatment. It is only used to activate the interface of the surface. However, there is a change in the size before and after metal coating. As shown in Figure 11, it can be seen that the 3.25um size is coated with a 133nm-thick metal.
11. Author said that “There was no significant difference compared with the commercially available products, and the contact resistance of the conductive balls” and “…there was no significant difference from the commercial products when they were coated with nickel or gold.” However, no information or SEM image of the commercial material is shown. I suggest that authors include these results performed with the commercial material, as well as the resistance or other properties informed by the manufacturer. Maybe the comparison between both materials could be shown in a table
-> When the commercially available product and the manufactured conductive ball are about 3um in size, the contact resistance is almost the same. However, by adding data when the sizes are 1um and 2um, we confirmed the appropriate size at the fine pitch. As a result, it has been added so that it can be compared with existing products.

Reviewer 2 Report
1. Introduction needs to be more comprehensive. Please include recent work that has already been done in the field and how your research adds value to this. Reading the introduction should give the reader an overview of ACFs apart from the specifics of the paper. Needs a lot more work.
2. References need to be more recent, 2016-2022.
3. In line 140, correct the symbol for specific gravity to be the same as in equation 1
4. Figure 6c spectra needs to be more clearer. Peak identification needs to be referenced properly. Kindly provide reference spectra, the value of the peaks mentioned in line 155 was increased from ? Use of spectra is unclear.
5. Separate out why spherical fabrication of polymers is important and why we need to coat them with metal ?
6. Why does removal of oxygen environment make the polymer spherical ? Mechanisms need to be more robust. illustrations can be useful.
7. Currently the paper is written as a lab report, kindly add some prior literature and build a theory around the two concepts of spherical polymer fabrications and metal coating them. WHat do you think is the underlying mechanism ? Its interesting why a oxygen deficit environment would create spherical polymer structure.
Author Response
to the reviewer
Thanks for the advice you pointed out.
I felt that there were a lot of things that needed to be corrected through the advice.
I added and corrected the missing data by referring to the advice.
The revised thesis will be sent again with attached files.
If you click Review in Word to check, you can check the revised part.
Thanks again.
------------------------------------------------------------------------------------
1. Introduction needs to be more comprehensive. Please include recent work that has already been done in the field and how your research adds value to this. Reading the introduction should give the reader an overview of ACFs apart from the specifics of the paper. Needs a lot more work.
-> The introduction has been completely revised. In addition, the mechanism and characteristic data you mentioned were additionally written.
2. References need to be more recent, 2016-2022.
-> An additional reference was made for the period you mentioned.
3. In line 140, correct the symbol for specific gravity to be the same as in equation 1
-> Modified to the same symbol.
4. Figure 6c spectra needs to be more clearer. Peak identification needs to be referenced properly. Kindly provide reference spectra, the value of the peaks mentioned in line 155 was increased from ? Use of spectra is unclear.
-> In the case of spectrum, only capture data exists because it is data measured by EDS. What I want to explain with this graph is to confirm that the surface is coated with nickel. When the surface image is mapped and measured, the inorganic substances appearing on the surface are displayed as peak values. As can be seen in Figure 6, the Ni peak value is the highest, so it can be seen that the surface is coated with Ni
5. Separate out why spherical fabrication of polymers is important and why we need to coat them with metal ?
-> Since the polymer is used, it is important to manufacture it in a spherical shape because the shape that can be coated most uniformly is a spherical shape. In addition, a surface metal coating is required because the circuit is connected to the circuit so that current can flow.
6. Why does removal of oxygen environment make the polymer spherical ? Mechanisms need to be more robust. illustrations can be useful.
-> When making a high molecular polymer, it is manufactured using a radical reaction. In this case, the radical polymerization mechanism can be explained by dividing it into four steps. These are the initiation, the propagation , the termination and the chain transfer. The propagation is a reaction in which monomers continue to bond and the chain length increases to form a polymer. At this time, when oxygen is present, radicals react with oxygen and the chain length is short, preventing the formation of polymers and causing a stop reaction. These contents were explained by adding a new mechanism part.
7. Currently the paper is written as a lab report, kindly add some prior literature and build a theory around the two concepts of spherical polymer fabrications and metal coating them. WHat do you think is the underlying mechanism ? Its interesting why a oxygen deficit environment would create spherical polymer structure.
-> The basic mechanism has been explained by adding a new part. Oxygen interferes with bonding when a radical reaction occurs, so proper polymer formation does not occur.

Round 2
Reviewer 1 Report
The authors were able to improve the manuscript, following the reviewers' suggestions. This made the manuscript clearer for the reader. I have just one last remark. The following sentence seems redundant (see "actual" and " actually "): “Figure 15 (a) is an actual image of the actually manufactured FPCB.” Based on this, I recommend that authors make a final revision of the manuscript text. And, I recommend publishing the article in Electronic Materials Journal.
Reviewer 2 Report
Additions look good.